# Intercomparison of online and offline XRF spectrometers for determining the

# 2 PM<sub>10</sub> elemental composition of ambient aerosol

- 3 Laura Cadeo<sup>1</sup>, Beatrice Biffi<sup>2</sup>, Benjamin Chazeau<sup>3,4</sup>, Cristina Colombi<sup>2</sup>, Rosario Cosenza<sup>2</sup>, Eleonora
- 4 Cuccia<sup>2</sup>, Manousos-Ioannis Manousakas<sup>3</sup>, Kaspar R. Daellenbach<sup>3,\*</sup>, André S. H. Prévôt<sup>3</sup>, Roberta
- 5 Vecchi<sup>1,\*\*</sup>

10

12

14

- <sup>1</sup>Department of Physics, Università degli Studi di Milano, Milan, 20133, Italy
- <sup>2</sup>ARPA Lombardia, Milan, 20124, Italy
- <sup>3</sup>Laboratory of Atmospheric Chemistry, Paul Scherrer Institute, 5232 Villigen PSI, Switzerland
- <sup>4</sup>Aix Marseille Univ, LCE, Marseille, France
- \*Corresponding authors: kaspar.daellenbach@psi.ch, roberta.vecchi@unimi.it
- **Keywords: X-ray fluorescence analysis;** XRF intercomparison, on-line spectrometry, PM<sub>10</sub>

#### 15 Abstract

Measuring the elemental composition of atmospheric particulate matter (PM) can provide useful 16 information on the adverse effects of PM and help the identification of emission sources. Carrying 17 out these measurements at a high time resolution (1-h or less) allows to describe the fast processes to which aerosol particles are subjected in the atmosphere, leading to a better characterisation of the 19 emissions. Energy dispersive X-ray fluorescence spectrometry (ED-XRF) is one of the most 20 widespread techniques used to determine the elemental composition of PM. In recent years, new 21 systems known as online XRF spectrometers have been developed to provide real-time measurements 22 of the PM elemental concentration at a high time resolution. Among these advanced instruments, the 23 Xact® 625i Ambient Metals Monitor by Cooper Environmental (USA) performs in situ automated 24 measurements with a user selected time resolution ranging from 15 to 240 min. In this study, an 25 Xact® 625i monitor equipped with a PM<sub>10</sub> inlet was deployed for nearly 6 months (July-December 26 2023) in Milan (Po Valley, Italy) at a monitoring station of the Lombardy Regional Agency for 27 Environmental Protection (ARPA Lombardia). The instrument was configured to quantify 36 28 elements, ranging from Al to Bi, with 1-h time resolution in the PM<sub>10</sub> fraction. The objective of the 29 study was to verify the correct functioning of the instrument and to evaluate the quality and robustness 30 of the data produced. Xact® 625i data were aggregated to 24-h daily means and then compared to 31 24-h PM<sub>10</sub> filter data retrieved by ARPA Lombardia in the same station and analyzed offline for the 32 elemental concentration with a benchtop ED-XRF spectrometer. The intercomparison focused on the 33 16 elements (Al, Si, S, Cl, K, Ca, Ti, Cr, Mn, Fe, Ni, Cu, Zn, Br, Sr, and Pb) whose concentrations 34 were consistently above their minimum detection limits (MDL) for both online and offline 35 techniques. Results of the intercomparison were satisfying showing that the Xact® 625i elemental 36

- concentrations were found to be highly correlated to the offline ED-XRF analyses ( $R^2$  ranging from
- 0.67 to 0.99) and slopes ranging from 0.79 to 1.3 (just a couple of elements showed slopes up to 1.70).

#### 1. Introduction

- Measurement and quantification of the chemical composition of atmospheric particulate matter (PM) 40 are key aspects of air quality monitoring. It has long been known that PM is associated with adverse 41 impacts, which are influenced by the chemical composition of the particles. At the global scale, PM 42 affects cloud formation and Earth's radiative budget (Fuzzi et al., 2015); at the local scale, its 43 harmfulness on human health is of particular concern (Brunekreef and Holgate, 2002; Kelly et al., 44 2012; Rohr and Wyzga, 2012; Daellenbach et al., 2020). Therefore, it is important to achieve a more 45 detailed knowledge about which chemical components are responsible for these negative effects. 46 Determining the composition of PM is also a fundamental step to perform source apportionment 47 studies for the identification of the emission sources, which help the implementation of mitigation 48 strategies (WHO, 2013). 49 Trace elements, in particular metals, although they generally do not contribute substantially to the 50 mass of PM are of interest because they act as tracers for specific sources (Visser et al., 2015) and 51 some of them are associated to adverse health effects even at ambient level concentrations (Chen and 52 Lippmann, 2009). The quantification of these elements in PM samples can be obtained through 53 various techniques (see e.g., Ogrizek et al., 2022), among the most widespread there are e.g., energy 54 dispersive X-ray fluorescence spectrometry (ED-XRF), particle-induced X-ray emission (PIXE), and 55
- the collection of aerosol particles on filters, followed by laboratory analysis. ED-XRF is a nondestructive technique and does not require any sample pre-treatment (e.g., repeated analyses on the

wet-chemistry inductively coupled plasma mass spectrometry (ICP-MS). All these methods require

- same sample and quantification of different chemical components in the same sample are possible),
- detects simultaneously multiple elements (20-30) with Z>10 using an X-ray tube for irradiating the
- samples, and it is typically operated using benchtop instruments. For decades until today, it has been
- largely applied to aerosol analysis in research laboratories as well as in monitoring networks like e.g.,
- the U.S. Environmental Protection Agency Chemical Speciation Network
- (https://www.epa.gov/amtic/chemical-speciation-network-csn-general-information). One advantage
- of ED-XRF is that it is quite stable and does not require frequent calibrations so that it is suitable for
- automated spectrometer development.
- PIXE analysis uses accelerated particles (often protons with energies of a few MeV) as irradiation
- source and it has been traditionally used to assess the elemental composition in aerosol filter samples
- (see e.g., Lucarelli et al., 2020; and therein cited literature). Although being more sensitive than ED-

XRF, PIXE has some features in common with ED-XRF such as the capability of providing 70 quantitative information for elements with Z>10 (being both based on fluorescence X-rays detection), 71 the unnecessary sample pre-treatment and the non-destructive character. While the need of an 72 accelerator facility makes the beam-time availability for PIXE analysis a shortcoming, the existence 73 74 of very effective PIXE set-up where a high number of filter samples can be robustly and effectively analyzed in short times helps a lot in large monitoring campaigns with many samples to be 75 characterized. As an example, at the INFN-LABEC facility in Florence, Italy, the typical irradiation 76 time for each daily aerosol sample is 45-60 s depending on the mass loading (vs. approximately 1-h 77 per sample with ED-XRF) and, more interestingly, also 1-h resolution samples can be analyzed in 1 78 min per spot (see e.g., Calzolai et al., 2010, 2015; Lucarelli et al., 2011). 79 ICP-MS is a very sensitive and fast analytical technique for detecting trace and ultra-trace elements 80 (>50 elements simultaneously) in aerosol samples (see e.g., Duarte et a., 2021); it is ideal for heavy 81 metals accurate quantification which is performed on solubilized samples by strong acid digestion 82 thus requiring a time-consuming step, introducing a dependence on the extraction efficiency and 83 possible sample contaminations, and destroying the filter sample. In addition, ICP-MS instruments 84 need frequent calibrations and strict quality control checks to ensure stable and robust element 85 detection. As far as aerosol source tracers are concerned, a major drawback of ICP-MS is the poor 86 detection of elements like Si which is a key tracer for mineral dust particles (see e.g., Canepari et al., 87 2009; Niu et al., 2010). 89 It is well-known that the ED-XRF technique is characterized by higher minimum detection limits (MDL) compared to ICP-MS (up to two orders of magnitude; see e.g., Hyslop et al., 2024) and PIXE 90 (up to one order of magnitude; see e.g., Calzolai et al., 2008); this is a limiting factor when very low 91 aerosol loadings or trace/ultra-trace elements are of interest but e.g., for source apportionment 92 purposes it proved to be effective also when analyzing sub-daily samples or size-segregated samples 93 (see e.g., Bernardoni et al., 2011a; Bernardoni et al., 2011b). The filter type used for the aerosol 94 samples also play a role in the technique performance as reported by previous literature works (see 95 e.g., Calzolai et al., 2008; Ogrizek et al., 2022). As far as low Z elements are concerned, especially 96 for heavy loaded samples (Hyslop et al., 2024), a limitation of techniques based on the detection of 97 fluorescence X-rays is the matrix effect, whereby emitted X-rays are reabsorbed by other particles in 98 the sample matrix or are self-absorbed within single coarse particles (Hunter, and Rhodes, 1972; Van Grieken and Markowicz, 1993) thus leading to an underestimation of the low-Z elemental 100 concentrations. However, these effects can be properly taken into account using correction factors 101 that can be either experimentally retrieved (see e.g., the use of PIGE-Particle Induced Gamma-ray 102

Emission analysis jointly with PIXE in Ariola et al., 2002; Calzolai et al., 2014) or by theoretical 103 calculations (see e.g., Hunter and Rhodes, 1972a, 1972b; Criss, 1976; Foster et al., 1996). 104 105 PM samples are usually collected by air quality monitoring networks with a time resolution ranging 106 from 24-h to one week, to ensure that enough PM mass is available for the analytical analysis, which 107 is carried out in a laboratory. The elemental composition of PM is then obtained with a considerable time delay and at low temporal resolution. In recent years, there has been a growing interest in 108 109 developing instruments for high temporal resolution measurements. Sampling at a high time resolution (1-h or less) allows to capture fast processes which aerosol particles are subjected to in the 110 atmosphere and to retrieve information about the typical hours of activity of a certain source, leading 111 to a better characterization of PM emissions. However, due to the short integration times, high-time 112 resolution measurements are often close to the MDL of the analytical techniques (Malaguti et al., 113 114 2015). Regarding the ED-XRF method, new systems have been developed which are able to sample PM 115 particles with a sub-hourly or hourly time resolution and to automatically measure their elemental 116 concentration, providing near-real time data access. These instruments are known as online XRF 117 spectrometers and can be employed for long monitoring periods (months, years) at a site with the 118 advantage of requiring limited maintenance. However, their high cost may prevent the simultaneous 119 120 use of multiple devices at different sites or the investigation of different size classes (Furger et al., 2017). One of these advanced instruments is the Xact® 625i Ambient Metals Monitor by Cooper 121 122 Environmental (USA), which performs in situ automated measurements of the elemental concentration of PM with a user selected time resolution ranging from 15 to 240 min. During 123 124 operation, remote access to the data is available, enabling continuous, near-real-time monitoring of the instrument and ambient metal concentrations. Although the Xact® is currently one of the most 125 126 widely used online ED-XRF analyzers, it is worth noting that other instruments with similar working principles are also available, such as the Horiba PX-375 ED-XRF monitor, whose setup and 127 performance are described in detail in Asano et al., (2017) and Trebs et al., (2024). 128 The Xact® 625i and its forerunner versions have been successfully employed in several field studies 129 in the past years, which compared its online measurements to daily samples analyzed with more 130 established laboratory techniques (Bhowmik et al., 2022; Tremper et al., 2018; Furger et al., 2017; 131 Park et al., 2014; US-EPA, 2012). Among these studies, only in Park et al., (2014) the daily filters 132 were analyzed with the ED-XRF technique; in all the other cases, the elemental concentration of daily 133 samples was retrieved with ICP-MS and ICP-OES (inductively coupled plasma optical emission 134 spectrometry). In the latest cases, the comparison was then influenced by the different choice of the 135 analytical technique. Moreover, in most of these studies, the experimental campaigns were carried 136

out only for a few weeks/months, leading to a very limited number of points available for the 137 intercomparison. An evaluation of the performances of Xact® 625i (compared with the ICP-MS 138 technique) during different seasons was conducted only by Bhowmik et al., (2022) who conducted 139 the field campaigns during summer (June-July) and winter (October-December) 2019 at two sites in 140 141 Delhi. In this study, an Xact® 625i monitor was deployed for nearly 6 months (July-December 2023) in 142 143 Milan (Po Valley, Italy) at a monitoring station of the Lombardy Regional Agency for Environmental Protection (ARPA Lombardia), where air quality measurements are performed continuously. Xact® 144 625i hourly samples measured online with ED-XRF were compared to daily filters measured offline 145 by ARPA Lombardia with a benchtop ED-XRF spectrometer in their laboratory. The goals of this 146 paper are (1) to assess the on-line instrument performance in typical summer and winter elemental 147 concentration ranges for PM<sub>10</sub> collected at an urban background site in the Po valley (Italy); (2) to 148 evaluate the quality of the obtained data for the selected elements in relation to their MDLs; (3) to 149 quantify the data robustness based on intercomparison between Xact® 625i and elemental 150 concentrations provided by a benchtop ED-XRF spectrometer. 151

#### 152 **2. Materials and methods**

#### 2.1 Site characteristics

153

The field campaign was performed at the permanent station Milano Pascal of the ARPA Lombardia 154 155 Air Quality Network from 6 July until 12 December 2023. This is an urban background site located in the eastern side of Milan, in the University campus area called "Città Studi" (45.478°N, 9.231°E; 156 122 m a.s.l); the station is placed in a public park about 130 m from road traffic. The metropolitan 157 city of Milan is the second most densely populated area in Italy (ca. 2300 inhabitants km<sup>-2</sup>, almost 158 doubled by daily commuters) and is located in the Po Valley, a well-known European pollution 159 hotspot. The site is characterized by wintertime episodes of high pollutant concentrations, due to 160 emissions from a variety of sources (e.g., residential heating, traffic, and industries) and prolonged 161 atmospheric stability conditions related to the presence of the mountain chains of the Alps and the 162 Apennines (Vecchi et al., 2007, 2009). Moreover, in Milan more than 80% of the days in a year are 163 characterized by wind speed lower than 2 m s<sup>-1</sup> (Vecchi et al., 2019). The site is well documented 164 with respect to gas-phase pollutants (e.g. NO<sub>x</sub>, SO<sub>2</sub>, O<sub>3</sub>), PM<sub>10</sub> and PM<sub>2.5</sub> chemical characterization, 165 and source apportionment (e.g., Amato et al., 2016; Altuwayjiri et al., 2021). 166

#### 2.2 Xact® 625i

167

The Xact® 625i Ambient Metals Monitor (Cooper Environmental Services (CES), Beaverton, OR, 168 USA) is an online energy dispersive XRF spectrometer, designed for continuous measurements of 169 the elemental composition of ambient aerosol. The device operates using a reel-to-reel filter tape 170 sampling technique, followed by the analysis of metals in the resulting PM spot through energy 171 dispersive X-ray fluorescence (ED-XRF). Ambient air is drawn inside the instrument through a PM 172 size-selective inlet which was PM<sub>10</sub> in this study, with a flow rate of 16.7 lpm and the PM is collected 173 onto a Teflon filter tape. After each sampling interval is completed, the filter tape is automatically 174 advanced to the XRF system, where the resulting PM deposit is irradiated with an X-ray tube 175 (rhodium anode, max voltage: 50 kV, current:1 mA) with three excitation conditions (see Table S1 176 in the Supplementary Material) and the fluorescence X-rays are measured by a silicon drift detector 177 178 (SDD). In the meantime, the next sample is collected on a clean spot of the filter tape and the process is repeated during each sampling interval, which was set at 60 min for this study. The XRF spectra 179 thus produced are automatically analyzed by a proprietary software for spectral analysis and 180 elemental quantification which is installed on the built-in computer. The software, through a linear 181 least-squares deconvolution algorithm, fits each measured spectrum with a library of pure element 182 reference spectra to obtain the concentration data for each calibrated element in ng m<sup>-3</sup>. Data can be 183 then downloaded and monitored remotely with an internet connection. Sampling and XRF analysis 184 are performed continuously and simultaneously, except for the time required for tape advancement 185 (~ 20 s). Quality assurance (QA) checks are performed every day at midnight for 30 min and consist 186 of an energy calibration (using a rod coated with Cr and Nb) and an upscale measurement to monitor 187 the stability of the instrument response (for Cr, Pb, and Cd). Therefore, the sample following midnight 188 189 is collected with a sampling interval limited to 30 min (00:30-01:00 LT). The instrument was located inside a temperature-controlled cabinet outside the ARPA Lombardia 190 191 monitoring station. If any errors are detected during operation, the system halts sampling, ramps the X-rays down for safety, and displays the cause of the error. The instrument was configured to quantify 192 193 36 elements: Al, Si, P, S, Cl, K, Ca, Ti, V, Cr, Mn, Fe, Co, Ni, Cu, Zn, Ga, Ge, As, Se, Br, Rb, Sr, Y, Zr, Cd, In, Sn, Sb, I, Ba, Hg, Tl, Pb, and Bi; in addition, Nb is also detected for daily QA checks. 194 Before the beginning of the experimental campaign, each of these elements was calibrated with a 195 reference standard sample. For each element, 1σ interference-free MDLs (MDL<sub>1σ</sub>) for 1-h of 196 sampling are reported in Table S2, provided for Xact® 625i following the approach reported in Currie 197 (1977). In XRF analyses, MDLs are inversely proportional to the square root of the irradiation time, 198 which in the case of Xact® 625i corresponds to the sampling interval. Therefore, lower MDLs are 199 reached for longer sampling durations. 200

## 2.3 Daily PM<sub>10</sub> filter samples

201

- Daily PM<sub>10</sub> samples were collected on mixed cellulose ester membrane filters (47 mm diameter) with
- a SWAM Dual Channel Monitor (FAI Instruments, Rome, Italy) equipped with PM<sub>10</sub> and PM<sub>2.5</sub> inlets.
- The elemental composition of PM samples was determined offline by ED-XRF spectrometry in the
- laboratories of the Environmental Monitoring Sector of ARPA Lombardia. An Epsilon 4 spectrometer
- from Malvern Panalytical (Monza, Italy) was used for the ED-XRF analysis. Four different irradiation
- conditions, which are reported in Table S3, were chosen to optimize the measurement of 19 elements,
- i.e., Al, Si, P, S, Cl, K, Ca, Ti, V, Cr, Mn, Fe, Ni, Cu, Zn, Br, Rb, Sr, and Pb. For these measurements,
- MDLs based on 24-h sampling time were evaluated as three times the square root of the counts in the
- background below the peak of the element divided by the corresponding sensitivity in the blank filter
- $(MDL_{3\sigma})$  (Jenkins, 1981; Lindgren, 2006) and are reported in Table S4.

## 212 **2.4 Data coverage**

- The Xact® 625i measurements started on 6 July 2023 16:00 LT (local time) and ended on 12
- December 2023 22:00 LT. The sampling interval of the instrument was set to 1-h. During the summer,
- in July and August, Xact® 625i suffered from high temperatures during the heatwaves, causing the
- X-ray tube to reach temperatures above 45° C. This led to automatic shutdowns of the measurements
- and to subsequent manual restarts, mostly performed remotely. The issue was mainly observed in the
- central hours of the day, from 13:00 to 16:00 LT. Nevertheless, it was still possible to attain a data
- coverage above 80% for Xact® 625i data in the central hours of the day during summertime. As a
- precautionary measure to avoid heat damage to the X-ray tube, Xact® 625i was switched off from 12
- to 24 August. During those days, a power failure in the ARPA Lombardia cabin caused also the
- interruption of daily measurements. Another power failure occurred from 22 October to 8 November,
- leading to a long pause of hourly measurements. The X-ray tube of Xact® 625i started malfunctioning
- on 6 December. The issue could not be resolved and the X-ray tube had to be replaced, resulting in a
- premature end of the experimental campaign.
- Overall, the Xact® 625i dataset consists of 2693 valid 1-h samples out of 3822 possible samples,
- attaining a coverage of 70%. For the daily filters, the dataset consists of 149 samples out of 157
- possible samples, reaching a coverage of 95%. A timeline of the periods in which data are missing is
- reported for both hourly and daily measurements in Figure S1. A summary of the periods of
- interruption of the measurements lasting more than 12 hours is reported in Table S5. The number N
- of overlapping days with validated data is reported in Table 1 for each element considered for the
- intercomparison.

- Xact® 625i data, which were originally reported in LT, were synchronized to daily samples and
- expressed in UTC+1 time zone.

#### 235 **2.5 Treatment of data below MDLs**

- Following the approach of Furger et al., (2017) and Tremper et al., (2018), for the intercomparison
- study here presented the MDL<sub>3 $\sigma$ </sub> was considered also for Xact® 625i; indeed, the MDL<sub>3 $\sigma$ </sub> assures a
- high statistical confidence (99.7%) and a better comparability with previous literature works.
- Hereafter,  $MDL_{3\sigma}$  will be referred simply to as MDL.
- All the elements measured on the daily filters by offline ED-XRF have less than 35% of their data
- points below their MDL. Among the elements detected by Xact® 625i, 13 of them (P, Co, Ga, Ge,
- Y, Cd, In, Sn, Sb, I, Hg, Tl, and Bi) have more than 90% of their data points below MDL; therefore,
- these elements were excluded from the intercomparison analysis. V and Rb have >70% of their data
- points below MDL leading to a less robust intercomparison with offline ED-XRF (see Figure S2). In
- Table S6, the number of data points with concentrations above the MDL is reported for each element
- measured by Xact® 625i and by offline ED-XRF.
- The intercomparison between Xact® 625i and daily PM<sub>10</sub> elemental concentrations was finally
- performed on 16 elements (Al, Si, S, Cl, K, Ca, Ti, Cr, Mn, Fe, Ni, Cu, Zn, Br, Sr, and Pb) which
- were measured by both techniques and were consistently above their MDLs.

#### 250 3. Results and discussion

## 251 **3.1 Data overview**

- An overview of the data recorded during the experimental campaign is given in Fig. 1, taking into
- account all available valid data of the elements considered for the intercomparison. To account for
- seasonal differences in terms of meteorology and emissions, data were divided into 3 periods: July-
- August, September-October and November-December. The basic statistics of the dataset, including
- the mean, median, standard deviations, 25<sup>th</sup> and 75<sup>th</sup> percentiles are reported in Table S7. As
- previously mentioned, the Xact® 625i data coverage for July and August was impacted by the loss
- of data mainly related to the time interval 13:00-16:00 LT when hot temperatures caused the X-ray
- tube switch-off; therefore, the statistical robustness of the comparison is lower than in the other
- represented periods.

**Figure 1:** Box plots of the concentrations for the elements considered for the intercomparison, measured hourly online (in red) and daily offline (in blue) during the experimental campaign in (a) July-August, (b) September-October, and (c) November-December. The bottom and the top of each box are the 25<sup>th</sup> and 75<sup>th</sup> percentiles, respectively; the line in the middle of the box is the median; the bottom and top whiskers are the minimum and maximum value respectively.

#### 3.2 Intercomparison data analysis approach

For the intercomparison between the two methods, Xact® 625i hourly data were averaged to 24-h to be comparable to the corresponding daily filter samples measured by offline ED-XRF. Every day, during QA checks performed from 00:00 to 00:30 LT, Xact® 625i generates one sample with a 30-

272 min time resolution so that this sample is added to the 23 hourly samples to calculate 24-h daily means. This procedure implicitly assumes that the half-hour sample collected during the first hour of 273 sampling is representative of the entire hour. The hypothesis was tested conducting 23.5 h weighted 274 means on a small number of samples, following the method of Furger et al., (2017). Tests showed a 275 276 difference of less than 3% between the 23.5 h weighted mean and the 24-h mean, which was then chosen as calculation method. For this reason, Xact® 625i data were aggregated to 24-h daily means. 277 278 As previously stated, during the campaign summer days were affected by heat waves, which caused 279 Xact® 625i to stop during the central hours of the day, leading to missing data. For this reason, the data coverage of Xact<sup>®</sup> 625i was evaluated for each day of the experimental campaign. In order to 280 avoid misestimation of daily Xact® 625i concentrations, days with less than 18 hourly valid data 281 (75% coverage) were excluded from the intercomparison. In addition, Xact® 625i daily means were 282 not calculated when more than 6 hourly data were under the MDL for one day. In all comparisons, 283 data under MDL were replaced by 0.5·MDL. 284 The comparisons between the daily PM<sub>10</sub> elemental concentrations retrieved by ARPA Lombardia 285 through offline ED-XRF and the daily means calculated from Xact® 625i hourly data were carried 286 out using the Deming regression (Deming, 1943). This regression approach minimizes the sum of 287 distances between the regression line and the X and Y variables, considering the experimental 288 uncertainties in both variables. For the offline ED-XRF measurements, the uncertainty included 289 contributions of 5% from calibration standard uncertainty (U.S. EPA, 1999) and, for each spectrum, 290 the contribution of counting statistics and fitting errors. For the Xact® 625i measurements, the 291 uncertainty included contributions of 5% from calibration standard uncertainty (U.S. EPA, 1999), 292 and an element-specific uncertainty derived from the spectral deconvolution calculated by the 293 instrument software for each spectrum, which includes also the contribution of the flow and the 294 295 sample deposit area. The mean relative uncertainties registered during the experimental campaign are reported for each element and for both online and offline methods in Table S8. 296

## 3.3 Intercomparison results

The results of the intercomparison between the  $PM_{10}$  elemental concentrations retrieved offline and online are reported in Table 1. The Deming regression parameters are reported along with their uncertainties and the coefficient of determination of the linear regression; the number of data (N) considered for the comparison after data reduction is also reported.

| Element | Slope ± uncertainty (online vs offline) | Intercept ± uncertainty [µg m <sup>-3</sup> ]<br>(online vs offline) | N   | $\mathbb{R}^2$ |
|---------|-----------------------------------------|----------------------------------------------------------------------|-----|----------------|
| Al      | $1.29 \pm 0.22$                         | $0.1640 \pm 0.0917$                                                  | 42  | 0.83           |
| Si      | $1.69 \pm 0.13$                         | $-0.2528 \pm 0.1003$                                                 | 97  | 0.94           |
| S       | $1.25 \pm 0.02$                         | $-0.0469 \pm 0.0070$                                                 | 101 | 0.99           |
| Cl      | $1.69 \pm 0.25$                         | $-0.0493 \pm 0.0290$                                                 | 75  | 0.67           |
| K       | $1.05 \pm 0.03$                         | $-0.0207 \pm 0.0054$                                                 | 102 | 0.97           |
| Ca      | $1.03 \pm 0.03$                         | $0.0038 \pm 0.0169$                                                  | 102 | 0.95           |
| Ti      | $1.00 \pm 0.06$                         | $-0.0006 \pm 0.0011$                                                 | 100 | 0.96           |
| Cr      | $1.30 \pm 0.08$                         | $0.0014 \pm 0.0004$                                                  | 77  | 0.86           |
| Mn      | $0.83 \pm 0.02$                         | $-0.0002 \pm 0.0003$                                                 | 101 | 0.95           |
| Fe      | $0.96 \pm 0.02$                         | $0.0385 \pm 0.0118$                                                  | 102 | 0.98           |
| Ni      | $0.79 \pm 0.06$                         | $0.0004 \pm 0.0001$                                                  | 79  | 0.87           |
| Cu      | $0.85 \pm 0.01$                         | $0.0006 \pm 0.0002$                                                  | 102 | 0.99           |
| Zn      | $0.98 \pm 0.02$                         | $-0.0001 \pm 0.0008$                                                 | 102 | 0.99           |
| Br      | $1.06 \pm 0.04$                         | $-0.0006 \pm 0.0002$                                                 | 96  | 0.96           |
| Sr      | $0.98 \pm 0.06$                         | $-0.0008 \pm 0.0002$                                                 | 74  | 0.97           |
| Pb      | $0.94 \pm 0.03$                         | $-0.0017 \pm 0.0003$                                                 | 83  | 0.99           |

**Table 1**: Deming regression results and coefficient of determination for the comparison between Xact® 625i (Y) and offline ED-XRF data (X). For each element, the number of points (N) available for the intercomparison is reported.

The scatterplots of the intercomparisons are presented in Figures 2-5. The time plots of the time series obtained by the two measurements methods are reported in Figures S3-S6. The 16 selected elements are compared by dividing them into three groups based on data characteristics.

The first group, Group A (Figures 2-3), includes K, Ca, Ti, Fe, Zn, Br, Sr, and Pb. This group shows excellent correlation between the two measurements methods ( $R^2 > 0.95$ ) and is characterized by slopes compatible to unity within three times the uncertainty of the fitted slope ( $3\sigma$ ). For Ca, Ti, and Zn also the intercepts are compatible to 0 within  $3\sigma$ . Among this group, K, Ca, Ti, Fe, and Zn, are measured by Xact® 625i with relative uncertainties in the range 10-20% (see Table S8). Br, Sr, and Pb are instead measured by Xact® 625i with a higher uncertainty, on average 30-50% (see Table S8), and Sr and Pb hourly data are also more frequently under the MDL (20% of data).

The second group, Group B (Figure 4), consists of the elements Si, S, Mn, and Cu. This group is characterized by excellent correlation between the two measurements methods ( $R^2 > 0.95$ ) but, in contrast to Group A, the slopes of the regressions are not compatible to 1 within  $3\sigma$ . Si and S are among the lightest elements measured by Xact® 625i and, along with Al, it can be tricky to measure with ED-XRF because of absorption effects due to the presence of air in the irradiation chamber (e.g. as typically occur in the XRF online measurements) and/or self-absorption inside the coarse particles

themselves (Hunter, and Rhodes, 1972; Van Grieken and Markowicz, 1993); these effects can lead to an underestimation of low-Z element concentrations. Nevertheless, looking at the results for Al, Si, and S, absorption effects seem not to be the cause of the observed discrepancy as Xact® 625i data are typically higher than offline ED-XRF analysis. Moreover, it should be noted that Si is detected by Xact® 625i with mean uncertainties of 30%, while S is detected with mean uncertainties of 10%. In the case of Mn and Cu, concentrations provided by Xact® 625i are constantly lower than daily offline measurements by approximately 15%.

**Figure 2**: Scatterplots of the intercomparison between Xact® 625i data and offline ED-XRF data for the elements K, Ca, Ti, and Fe of Group A. The error bars represent the mean experimental uncertainties reported in Table S8.

**Figure 3**: Scatterplots of the intercomparison between Xact® 625i data and offline ED-XRF data for the elements Zn, Br, Sr, and Pb of Group A. The error bars represent the mean experimental uncertainties reported in Table S8.

**Figure 4**: Scatterplots of the intercomparison between Xact® 625i data and offline ED-XRF data for the elements of Group B: Si, S, Mn, and Cu. The error bars represent the mean experimental uncertainties reported in Table S8.

A possible explanation for the observed discrepancies is related to the fact that, despite all samples are measured through ED-XRF technique, the spectra analysis for quantitative analysis is different and — more importantly - the two instruments are not calibrated with the same set of certified standards, which can lead to different quantification of concentrations. However, Xact® 625i data of the elements of this group can still be validated when compared to an offline measurement technique and used for high-time resolution elemental concentrations assessment, after harmonisation of the datasets.

**Figure 5**: Scatterplots of the intercomparison between Xact® 625i data and offline ED-XRF data for the elements of Group C: Al, Cl, Cr, and Ni. The error bars represent the mean experimental uncertainties reported in Table S8.

The third group of elements, Group C (Figure 5), is composed of Al, Cl, Cr, and Ni. This group shows less comparability between the two methods, with  $R^2$  in the range 0.67-0.87. Cl, Cr, and Ni are frequently close or under the MDL for both experimental techniques and are characterized by mean relative uncertainties in the range of 30-50%. For these elements, the comparison could be improved by carrying out the Xact® 625i measurements on a 2 h time scale. Among the 16 elements evaluated for the intercomparison, Al is the one with the highest MDL for Xact® 625i hourly measurements and its hourly concentrations are under the MDL for nearly 35% of data points, while Al offline data are always above the MDL. Al is also measured by Xact® 625i with mean uncertainties of 50%. As can be seen also in Figure S6a, the Xact® 625i time series of Al is characterized by a constant upward shift in background concentrations, which is not observed for the other elements. The measurement

of Al with Xact® 625i is complicated by the fact that the instrument uses an Al filter to carry out the analysis, as reported in Table S1; another possible issue could be that the X-rays hit some internal 381 parts of the instrument, causing a significant increase of the background. Al concentrations cannot 382 383 thus be corrected in a reliable way and further improvements in the instrument should be considered 384 to enhance Al detection. In the case of Cl, which shows quite scattered data, concentrations obtained by Xact® 625i are on average higher than the ones measured offline on daily samples. This could be 385 386 explained by the volatility of Cl. Xact® 625i ED-XRF measurements are performed immediately 387 after the collection of the sample, while daily PM<sub>10</sub> filters are stored in the sampler at the monitoring station for up to 2 weeks before being taken to the laboratory for the offline ED-XRF analysis. 388 The results of this study represent a significant step forward from Park et al., (2014), which is – as 389 far as we know - the only previous study available in literature presenting a comparison between 390 391 Xact® hourly data and offline ED-XRF daily data. Park et al., (2014) conducted an experimental campaign with a forerunner version of Xact® (Xact® 620) in Gwangju, South Korea. The campaign 392 was carried out during February 2011 and lasted only 1 month, focusing on the PM<sub>2.5</sub> fraction. The 393 Xact® 620 model, was the first commercially available near real time ambient metals monitor; it was 394 able to detect elements starting from K and had higher detection limits (details can be found in Park 395 et al., 2014), required more manual intervention for calibration and quality assurance processes and 396 had a more basic interface with limited remote access capabilities. The daily filters were measured 397 offline with an Epsilon 5 ED-XRF spectrometer (Malvern Panalytical). The study compared the 398 399 online and offline concentrations of 12 elements (K, Ca, Ti, V, Mn, Fe, Ni, Cu, Zn, As, Ba, Pb), 9 of which were also analyzed in our study. For the 9 common elements (K, Ca, Ti, Mn, Fe, Ni, Cu, Zn, 400 Pb), they observed a mean  $R^2$  of 0.89 and a slope of 1.31, with Xact® measurements on average 30% 401 higher than offline ED-XRF. In our study, for these 9 elements, we found a much better correlation, 402 with a mean  $R^2$  of 0.96 and slope of 0.94, which is closer to unity. Moreover, our study included also 403 7 elements (Al, Si, S, Cl, Cr, Br, Sr) which were not taken into account by Park et al., (2014), and the 404 measurement campaign lasted for a longer period (6 months), giving more robustness to the results. 405 Overall, considering all the 16 elements evaluated in this study, we found a good correlation (mean 406  $R^2$  of 0.93) between the online and offline ED-XRF, with a mean slope of 1.11. The results are also 407 in agreement with Tremper et al. (2018), which compared Xact® measurements to ICP-MS daily 408 measurements in three sites in the United Kingdom. They observed a mean  $R^2$  of 0.93 and a slope of 409 1.07 for the elements As, Ba, Ca, Cr, Cu, Fe, K, Mn, Ni, Pb, Se, Sr, Ti, V, and Zn. In the study by 410 Furger et al. (2017), they found instead that the elemental measurements by an Xact® 625i were on 411 average 28% higher than ICP-OES and ICP-MS measurements for S, K, Ca, Ti, Mn, Fe, Cu, Zn, Ba, 412 and Pb. 413

A summary of previous literature studies and their characteristics is reported in Tables S9 - S10. In 414 these studies, several reasons for the differences observed between the Xact® data and the offline 415 techniques are described and some of them are shortly reported below. In general, as specified in 416 Tremper et al., (2018), the measured elements are chosen to represent a range of source categories 417 418 (i.e. regulatory, traffic, industry), plus the internal standard (Pd for Xact® 625 and Nb for Xact® 625i). The number of elements that are actually quantified and thus included in the intercomparison 419 420 results for each study depends on the ambient air concentrations, and thus the site, and MDL of the techniques. 421 In the US-EPA (2012) work, intercomparison results are available only for 6 elements, as the others 422 were under the MDL of ICP-MS analysis and/or Xact® measurements; weak regression parameters 423 for Cu are explained by concentrations frequently close to MDL of both techniques. In Furger et al., 424 (2017), Xact® and ICP-MS data showed high linearity and little scatter in the regressions for the 425 elements S, K, Ca, Ti, Mn, Fe, Cu, Zn, Ba, and Pb; the relative mean difference of 28% they have 426 found was attributed to many possible causes such as differences in the inlets used for the Xact® and 427 the high-volume samplers for ICP-MS filter samples, a slightly different location of the samplers, 428 possible calibration issues with the Xact® for S, values next to MDLs for one or both techniques, 429 XRF particle-size-dependent self-absorption effects for the lighter elements, and line interferences or 430 contaminations during the ICP-MS digestion and analysis procedures. Tremper et al. (2018) and 431 Bhowmik et al. (2022) both mentioned similar reasons for the differences observed between Xact® 432 433 and ICP-MS filter data; in addition, blank filters were found to be variable, the standards used for Xact® calibration had a much higher concentration than ambient air and the calibration matrix 434 435 differed from the sample matrix.

#### 4 Conclusions

This study was realized to evaluate the performances of an Xact® 625i online energy dispersive XRF 437 spectrometer. Although X-ray fluorescence is notably less sensitive than other analytical techniques 438 like ICP-MS, it is robust and stable so that online spectrometers can be deployed also in monitoring 439 networks due to easy use and little maintenance. Online spectrometers are still quite expensive and 440 only a reduced number of elements are detectable compared to e.g., ICP-MS but for source 441 apportionment studies the availability of high resolution elemental composition is currently key to 442 refined modelling applications. Indeed, the possibility of joining high-time resolution, which provide 443 details on temporal patterns, and low-time resolution elemental data, which allow the detection of 444 elemental tracers for specific sources, has been already proved to be effective for source 445 apportionment studies (see e.g., Crespi et la., 2016; Forello et al., 2019; Mooibroek et al, 2022). 446

A six-month experimental campaign was carried out at the ARPA Lombardia monitoring station Milano Pascal (Milan, Italy) from July to December 2023. The instrument was configured to 448 449 continuously measure 36 elements, ranging from Al to Bi, with 1-h time resolution. The measurement quality of Xact® 625i was tested by intercomparison with ED-XRF offline analyses on 24-h PM<sub>10</sub> 450 451 samples with a well-established benchtop spectrometer. Xact® 625i hourly data were aggregated to 24-h means and compared to daily PM<sub>10</sub> data. The study focused on 16 elements which were measured 452 453 by both techniques and were consistently above their MDLs (Al, Si, S, Cl, K, Ca, Ti, Cr, Mn, Fe, Ni, Cu, Zn, Br, Sr, and Pb). 454 Xact® 625i was found to be a highly reliable instrument, suitable for measurements of elemental 455 concentration of PM<sub>10</sub> in summer and winter conditions at 1-h time resolution. Xact® 625i elemental 456 concentrations were found to be highly correlated to the offline ED-XRF analyses of the daily samples 457  $(R^2 \text{ in the range } 0.67\text{-}0.99)$  albeit with slopes ranging from 0.79 to 1.70. Elements were divided into 458 three groups according to their characteristics. The first group, Group A (K, Ca, Ti, Fe, Zn, Br, Sr, 459 and Pb), shows excellent correlation between the two measurements methods ( $R^2 > 0.95$ ) and slopes 460 compatible to 1 (range 0.94-1.06). Group B (Si, S, Mn, and Cu) is still characterized by excellent 461 correlations between the two techniques, but the regression slopes are not compatible to 1. Xact® 462 625i performances are more critical for the elements of Group C (Al, Cl, Cr, Ni). These elements are 463 frequently under the MDL for one or both experimental techniques and show the worst correlations 464 between the two methods ( $R^2$  ranging from 0.67 to 0.87). An issue of the Xact® 625i instrument is 465 related to the quantification of Al, which is problematic so that the Al concentrations are basically 466 not reliable. 467 468 Future work should include an intercomparison between an Xact® 625i and an offline ED-XRF spectrometer calibrated with the same certified standards, in order to avoid biases linked to the 469 calibration of the instruments. Moreover, it would be interesting to assess the reliability of Xact® 470 625i high time resolution measurements by comparing it to other instruments/technique able to 471 perform measurements of PM elemental concentration at high time resolution, like Horiba PX-375 472

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

# 631 Author contribution

639

642

646

- **LC**: Data curation, Formal analysis, Visualization, Investigation, Writing original draft preparation,
- Writing review & editing; **BB**: Data curation, Investigation, Writing review & editing; **BC**: Data
- curation, Writing review & editing; CC: Resources, Project administration, Writing review &
- editing; **RC**: Investigation; **EC**: Investigation, Data curation, Validation; Writing review & editing;
- MIM: Data curation, Investigation, Writing review & editing; KRD: Resources; Validation;
- Writing review & editing; **ASHP**: Project administration, Resources, Writing review & editing;
- **RV**: Supervision, Validation, Writing review & editing.

### 640 Competing interests

The authors declare that they have no conflict of interest.

#### 643 Acknowledgements

- The Department of Physics of the University of Milan is acknowledged for the fellowship provided
- to Laura Cadeo.