# Peer review of "Intercomparison of online and offline XRF spectrometers for determining the"

_EGUsphere, 2025_

## Referee Comment (RC2)

Measuring the elemental composition of atmospheric particulate matter (PM) in high resolution is of interest to better identify emission sources. Cadeo et al. conducted a 6-month measurement to evaluate the performance and data quality of an online Xact® 625i trace elements monitor. This is good work with novel results and logical writing, and is recommended to be published in AMT after addressing the following problems.

**Major comments:**

1. Lines 318-321: Please discuss the instrument limitation, especially for Al. If the extremely high concentration of Al measured by the online instrument was likely caused by the "Al filter", the Xact® 625i should not be used to measure Al. At least, the authors should clarify that if it is possible to develop a correction method to eliminate the constant upward background influence.

2. Lines 327-337: Please explain what is the difference between the forerunner version of Xact (Xact 620) used by Park et al. and Xact® 625i, leading to a better measurement performance in the present study.

**Minor suggestion:**

1. Suggest changing the word "realized" in lines 99 and 350 to "conducted".

2. Please follow Figure 2, adding element names to the subplots in Figures 3 and 4.

---

## Author Comment (AC1)

*We thank the Reviewers and the Editor for their positive evaluation of this work and for their useful comments. Our replies to each comment are reported below and refer to the file "Cadeo et al_REV_tracked", where all changes to the original manuscript are visible.*

**Reviewer #1**: Online chemical characterisation of atmospheric particulate matter (PM) is a topic of interest for Air Quality and atmospheric research. In the recent years there are networks for atmospheric research implementing a combination of ACSM / XRF online / Aethalometers for a complete online characterisation of PM. Instruments for continuous analysis of metals in PM are based on EDXRF measurement. Nowadays, the Xact625i is the most widely used. However, these measurements have limitations mainly related to the measurement technique and the low concentration of PM sampled during short periods.

This paper evaluates the performance of the Xact625i, measuring PM10 with hourly resolution, by comparing with the offline analysis of PM collected in filters during 24h and analyzed offline by EDXRF. Novelty of the manuscript is the comparison with offline EXRF or a long time period (6 months). Results obtained are quite good for most selected elements. Three groups of elements have been identified base on different regression coefficients and slopes. Reasons of these differences are not properly discussed.

Overall, results obtained are of interest and merit publication. However, a major discussion about limitations of the technique and comparison with previous studies is needed.

Line 26: please, indicate that the Xact was equipped with a PM10 size cut inlet

*Done, see line 26.*

Line 60. You should also comment about limitations of XRF: detection limits of some tracers of interest can be too high compared with PIXE or ICP-MS analyses. Also. Analysis depends on the matrix; so, it can vary depending on the composition of the aerosol.

*We agree, this point could be stressed more in the paper. In the Introduction of the revised version (see lines 53-104) we addressed the main differences among the techniques. All of them have pros and cons and a brief summary of the instruments' main features is provided.*

Section 2.4. What was the minimum number of hours of valid data considered to calculate daily averages for the intercomparison?

*As reported in lines 320-325 of Section 3.2, in order to calculate daily averages from Xact® 625i hourly data, days with less than 18 hourly valid data (75% coverage) were excluded from the intercomparison. Moreover, Xact® 625i daily means were not calculated for days during which more than 6 hourly data were under the MDL for a certain element.*

Figure 1. Considering all available measurements or only the simultaneous ones?

*The box plots in Figure 1 were obtained by considering all available valid data of the elements considered for the intercomparison. We added this clarification in lines 252-253.*

Is there any difference in the correlations considering the three periods separately?

*We consider the comparison among all available data more robust than the one considering only sub-sets due to the higher statistical significance and the possibility of comparing the results over a broader concentration range. Indeed, as shown in Figure S1, when the three periods are considered separately the availability of parallel data is largely reduced and intermittent data availability for some elements (see e.g., group C in Figure S6)- due to instrument failure - makes the comparison quite puzzling. This is especially evident for July-August data when – as reported in Section 2.4 - heatwaves caused the X-ray tube to reach temperatures above 45° C. This led to automatic shutdowns of the measurements and to subsequent manual restarts, mostly performed remotely. The issue was mainly observed in the central hours of the day, from 13:00 to 16:00 LT. Therefore, these data have been included in the general dataset but the comparison with 24h filter data is less reliable.*

*We now pointed out this issue also in Section 3.1 (see lines 257-260) so that the reader is aware of the different reliability of the represented data.*

The discussion about comparison with previous studies should be improved. Are all the differences attributable to the calibration? Can be related to the local PM composition? What is the explanation for the different elements selected at each study? The MDL / ambient concentrations?

*As suggested by the Reviewer, we improved the discussion regarding comparisons with previous studies (see Section 3.3, lines 414-435, in the revised version). Indeed, two tables summarizing previous literature works dealing with the same intercomparison exercise are now reported in the Supplementary Material. Information on the type of technique used for offline measurements, the Xact® model, the time resolution of Xact® measurements, the elements measured offline, the elements measured online by Xact®, and the site, period and season of the measurement campaigns are reported in Table S9. In Table S10, a summary of the regression parameters (slope and R2) obtained in the previous literature studies is reported, compared to the results of our study.*

Different sizes may affect the comparison for some elements given the matrix effect. Park et al., used a forerunner version equipped with a PM2.5 inlet. Which inlet size was used by Tremper et al., 2018? And by Furger et al., 2017?

*We agree, the matrix effect is one of the limitations of the XRF technique, whereby emitted X-rays are reabsorbed by other particles in the sample matrix. This issue has been addressed in the revised text. The size of the inlets used in previous literature studies are reported in Table S9.*

Conclusions: you should add some statements about limitations of the technique. For example, limitations for the analysis of some key tracers such as V, Ba, Sn, As...

*Done, see the revised version, lines 438-446.*

Line 370: You should add that this Al measurement limitation is related to the instrument.

*Done, see the revised version, lines 465-467.*

---

## Author Comment (AC2)

*We thank the Reviewers and the Editor for their positive evaluation of this work and for their useful comments. Our replies to each comment are reported below and refer to the file "Cadeo et al_REV_tracked", where all changes to the original manuscript are visible.*

**Reviewer #2:** Measuring the elemental composition of atmospheric particulate matter (PM) in high resolution is of interest to better identify emission sources. Cadeo et al. conducted a 6- month measurement to evaluate the performance and data quality of an online Xact® 625i trace elements monitor. This is good work with novel results and logical writing, and is recommended to be published in AMT after addressing the following problems.

Major comments:
1. Lines 318-321: Please discuss the instrument limitation, especially for Al. If the extremely high concentration of Al measured by the online instrument was likely caused by the "Al filter", the Xact® 625i should not be used to measure Al. At least, the authors should clarify that if it is possible to develop a correction method to eliminate the constant upward background influence.

*Done, see the revised version of the paper, see lines 377-384 and 465-467.*

2. Lines 327-337: Please explain what is the difference between the forerunner version of Xact (Xact 620) used by Park et al. and Xact® 625i, leading to a better measurement performance in the present study.

*Done, see the revised version of the paper, lines 393-397.*

Minor suggestion:
1. Suggest changing the word "realized" in lines 99 and 350 to "conducted".

*Done.*

2. Please follow Figure 2, adding element names to the subplots in Figures 3 and 4.

*Done, see Figures 2-5.*